Too trivial to test? An inverse view on defect prediction to identify methods with low fault risk

http://orcid.org/0000-0002-9563-6743 Niedermayr Rainer 1 2 niedermayr@cqse.eu
Röhm Tobias 1
http://orcid.org/0000-0002-5256-8429 Wagner Stefan 2
1 CQSE GmbH , München , Germany
2 Institute of Software Technology, University of Stuttgart , Stuttgart , Germany
Bernardi Mario Luca
Electronic publication date: 2019 Apr 15
Publication date: 2019
Volume: 5
Electronic Location ID: e187
Received 2018 Oct 25; Accepted 2019 Mar 18
Copyright: © 2019 Niedermayr et al.
Copyright year: 2019
Copyright holder: Niedermayr et al.
License: This is an open access article distributed under the terms of the Creative Commons Attribution License, which permits unrestricted use, distribution, reproduction and adaptation in any medium and for any purpose provided that it is properly attributed. For attribution, the original author(s), title, publication source (PeerJ Computer Science) and either DOI or URL of the article must be cited.
License URL: https://creativecommons.org/licenses/by/4.0/

Keywords: Testing, Inverse defect prediction, Fault risk, Low-fault-risk methods

Funding: Institute of Software Technology of the University of Stuttgart German Federal Ministry of Education and Research (BMBF) SOFIE, 01IS18012A This work was funded by the Institute of Software Technology of the University of Stuttgart and the German Federal Ministry of Education and Research (BMBF), grant “SOFIE, 01IS18012A.” The funders had no role in study design, data collection and analysis, decision to publish, or preparation of the manuscript.

==============================
Background

Test resources are usually limited and therefore it is often not possible to completely test an application before a release. To cope with the problem of scarce resources, development teams can apply defect prediction to identify fault-prone code regions. However, defect prediction tends to low precision in cross-project prediction scenarios.

Aims

We take an inverse view on defect prediction and aim to identify methods that can be deferred when testing because they contain hardly any faults due to their code being “trivial”. We expect that characteristics of such methods might be project-independent, so that our approach could improve cross-project predictions.

Method

We compute code metrics and apply association rule mining to create rules for identifying methods with low fault risk (LFR). We conduct an empirical study to assess our approach with six Java open-source projects containing precise fault data at the method level.

Results

Our results show that inverse defect prediction can identify approx. 32–44% of the methods of a project to have a LFR; on average, they are about six times less likely to contain a fault than other methods. In cross-project predictions with larger, more diversified training sets, identified methods are even 11 times less likely to contain a fault.

Conclusions

Inverse defect prediction supports the efficient allocation of test resources by identifying methods that can be treated with less priority in testing activities and is well applicable in cross-project prediction scenarios.

Introduction

In a perfect world, it would be possible to completely test every new version of a software application before it was deployed into production. In practice, however, software development teams often face a problem of scarce test resources. Developers are busy implementing features and bug fixes, and may lack time to develop enough automated unit tests to comprehensively test new code (Ostrand, Weyuker & Bell, 2005; Menzies & Di Stefano, 2004). Furthermore, testing is costly and, depending on the criticality of a system, it may not be cost-effective to expend equal test effort to all components (Zhang, Zhang & Gu, 2007). Hence, development teams need to prioritize and limit their testing scope by restricting the code regions to be tested (Menzies et al., 2003; Bertolino, 2007). To cope with the problem of scarce test resources, development teams aim to test code regions that have the best cost-benefit ratio regarding fault detection. To support development teams in this activity, defect prediction has been developed and studied extensively in the last decades (Hall et al., 2012; D’Ambros, Lanza & Robbes, 2012; Catal, 2011). Defect prediction identifies code regions that are likely to contain a fault and should therefore be tested (Menzies, Greenwald & Frank, 2007; Weyuker & Ostrand, 2008).

This paper suggests, implements, and evaluates another view on defect prediction: inverse defect prediction (IDP). The idea behind IDP is to identify code artifacts (e.g., methods) that are so trivial that they contain hardly any faults and thus can be deferred or ignored in testing. Like traditional defect prediction, IDP also uses a set of metrics that characterize artifacts, applies transformations to pre-process metrics, and uses a machine-learning classifier to build a prediction model. The difference rather lies in the predicted classes. While defect prediction classifies an artifact either as buggy or non-buggy, IDP identifies methods that exhibit a low fault risk (LFR) with high certainty and does not make an assumption about the remaining methods, for which the fault risk is at least medium or cannot be reliably determined. As a consequence, the objective of the prediction also differs. Defect prediction aims to achieve a high recall, such that as many faults as possible can be detected, and a high precision, such that only few false positives occur. In contrast, IDP aims to achieve high precision to ensure that LFR methods contain indeed hardly any faults, but it does not necessarily seek to predict all non-faulty methods. Still, it is desired that IDP achieves a sufficiently high recall such that a reasonable reduction potential arises when treating LFR methods with a lower priority in QA activities.

Research goal: We want to study whether IDP can reliably identify code regions that exhibit only a LFR, whether ignoring such code regions—as done silently in defect prediction—is a good idea, and whether IDP can be used in cross-project predictions.

To implement IDP, we calculated code metrics for each method of a code base and trained a classifier for methods with LFR using association rule mining. To evaluate IDP, we performed an empirical study with the Defects4J dataset (Just, Jalali & Ernst, 2014) consisting of real faults from six open-source projects. We applied static code analysis and classifier learning on these code bases and evaluated the results. We hypothesize that IDP can be used to pragmatically address the problem of scarce test resources. More specifically, we hypothesize that a generalized IDP model can be used to identify code regions that can be deferred when writing automated tests if none yet exist, as is the situation for many legacy code bases.

Contributions: (1) The idea of an inverse view on defect prediction: While defect prediction has been studied extensively in the last decades, it has always been employed to identify code regions with high fault risk. To the best of our knowledge, the present paper is the first to study the identification of code regions with LFR explicitly. (2) An empirical study about the performance of IDP on real open-source code bases. (3) An extension to the Defects4J dataset (Just, Jalali & Ernst, 2014): To improve data quality and enable further research—reproduction in particular—we provide code metrics for all methods in the code bases and an indication whether they were changed in a bug-fix patch, a list of methods that changed in bug fixes only to preserve API compatibility, and association rules to identify LFR methods.

The remainder of this paper is organized as follows. “Association Rule Mining” provides background information about association rule mining. “Related Work” discusses related work. “IDP Approach” describes the IDP approach, that is, the computation of the metrics for each method, the data pre-processing, and the association rule mining to identify methods with LFR. Afterward, “Empirical Study” summarizes the design and results of the IDP study with the Defects4J dataset. Then, “Discussion” discusses the study’s results, implications, and threats to validity. Finally, “Conclusion” summarizes the main findings and sketches future work.

Association Rule Mining

Association rule mining is a technique for identifying relations between variables in a large dataset and was introduced by Agrawal, Imieliński & Swami (1993). A dataset contains transactions consisting of a set of items that are binary attributes. An association rule represents a logical implication of the form {antecedent} → {consequent} and expresses that the consequent is likely to apply if the antecedent applies. Antecedent and consequent both consist of a set of items and are disjoint. The support of a rule expresses the proportion of the transactions that contain both antecedent and consequent out of all transactions. The support of an item X with respect to all transactions T is defined as supp(X)=|t∈T:X⊆t||T|. It is related to the significance of the itemset (Simon, Kumar & Li, 2011). The confidence of a rule expresses the proportion of the transactions that contain both antecedent and consequent out of all transactions that contain the antecedent. The confidence of a rule X → Y is defined as conf(X→Y)=supp(X∪Y)supp(X). It can be considered as the precision (Simon, Kumar & Li, 2011). A rule is redundant if a more general rule with the same or a higher confidence value exists (Bayardo, Agrawal & Gunopulos, 1999).

Association Rule Mining has been successfully applied in defect prediction studies (Song et al., 2006; Czibula, Marian & Czibula, 2014; Ma et al., 2010; Zafar et al., 2012). A major advantage of association rule mining is the natural comprehensibility of the rules (Simon, Kumar & Li, 2011). Other commonly used machine-learning algorithms for defect prediction, such as support vector machines or Naive Bayes classifiers, generate black-box models, which lack interpretability. Even decision trees can be difficult to interpret due to the subtree-replication problem (Simon, Kumar & Li, 2011). Another advantage of association rule mining is that the gained rules implicitly extract high-order interactions among the predictors.

Related Work

Defect prediction is an important research area that has been extensively studied (Hall et al., 2012; Catal & Diri, 2009). Defect prediction models use code metrics (Menzies, Greenwald & Frank, 2007; Nagappan, Ball & Zeller, 2006; D’Ambros, Lanza & Robbes, 2012; Zimmermann, Premraj & Zeller, 2007), change metrics (Nagappan & Ball, 2005; Hassan, 2009; Kim et al., 2007), or a variety of further metrics (such as code ownership (Bird et al., 2011; Rahman & Devanbu, 2011), developer interactions (Meneely et al., 2008; Lee et al., 2011), dependencies to binaries (Zimmermann & Nagappan, 2008), mutants (Bowes et al., 2016), code smells (Palomba et al., 2016) to predict code areas that are especially defect-prone. Such models allow software engineers to focus quality-assurance efforts on these areas and thereby support a more efficient resource allocation (Menzies, Greenwald & Frank, 2007; Weyuker & Ostrand, 2008).

Defect prediction is usually performed at the component, package or file level (Nagappan & Ball, 2005; Nagappan, Ball & Zeller, 2006; Bacchelli, D’Ambros & Lanza, 2010; Scanniello et al., 2013). Recently, more fine-grained prediction models have been proposed to narrow down the scope for quality-assurance activities. Kim, Whitehead & Zhang (2008) presented a model to classify software changes. Hata, Mizuno & Kikuno (2012) applied defect prediction at the method level and showed that fine-grained prediction outperforms coarse-grained prediction at the file or package level if efforts to find the faults are considered. Giger et al. (2012) also investigated prediction models at the method level and concluded that a Random Forest model operating on change metrics can achieve good performance. More recently, Pascarella, Palomba & Bacchelli (2018) replicated this study and confirmed the results. However, they reported that a more realistic inter-release evaluation of the models shows a dramatic drop in performance with results close to that of a random classifier and concluded that method-level bug prediction is still an open challenge (Pascarella, Palomba & Bacchelli, 2018). It is considered difficult to achieve sufficiently good data quality at the method level (Hata, Mizuno & Kikuno, 2012; Shippey et al., 2016); publicly available datasets have been provided in Shippey et al. (2016), Just, Jalali & Ernst (2014), and Giger et al. (2012).

Cross-project defect prediction predicts defects in projects for which no historical data exists by using models trained on data of other projects (Zimmermann et al., 2009; Xia et al., 2016). He et al. (2012) investigated the usability of cross-project defect prediction. They reported that cross-project defect prediction works only in few cases and requires careful selection of training data. Zimmermann et al. (2009) also provided empirical evidence that cross-project prediction is a serious problem. They stated that projects in the same domain cannot be used to build accurate prediction models without quantifying, understanding, and evaluating process, data and domain. Similar findings were obtained by Turhan et al. (2009), who investigated the use of cross-company data for building prediction models. They found that models using cross-company data can only be “useful in extreme cases such as mission-critical projects, where the cost of false alarms can be afforded” and suggested using within-company data if available. While some recent studies reported advances in cross-project defect prediction (Xia et al., 2016; Zhang et al., 2016; Xu et al., 2018), it is still considered as a challenging task.

Our work differs from the above-mentioned work in the target setting: we do not predict artifacts that are fault-prone, but instead identify artifacts (methods) that are very unlikely to contain any faults. While defect prediction aims to detect as many faults as possible (without too many false positives), and thus strives for a high recall (Mende & Koschke, 2009), our IDP approach strives to identify those methods that are not fault-prone to a high certainty. Therefore, we optimized our approach toward the precision in detecting LFR methods. To the best of our knowledge, this is the first work to study LFR methods. Moreover, as far as we know, cross-project prediction has not yet been applied at the method level. To perform the classification, we applied association rule mining. Association rule mining has previously been applied with success in defect prediction (Song et al., 2006; Morisaki et al., 2007; Czibula, Marian & Czibula, 2014; Ma et al., 2010; Karthik & Manikandan, 2010; Zafar et al., 2012).

IDP Approach

This section describes the IDP approach, which identifies LFR methods. The approach comprises the computation of source-code metrics for each method, the data pre-processing before the mining, and the association rule mining. Figure 1 illustrates the steps.

Figure 1 Overview of the approach.

Metrics for faulty methods are computed at the faulty state; metrics for non-faulty methods are computed at the state of the last bug-fix commit.

Metric computation

Like defect prediction models, IDP uses metrics to train a classifier for identifying LFR methods. For each method, we compute the source-code metrics listed in Table 1 that we considered relevant to judge whether a method is trivial. They comprise established length and complexity metrics used in defect prediction, metrics regarding occurrences of programming-language constructs, and categories describing the purpose of a method.

Table 1 Computed metrics for each method.

	Metric name	Type	
M1	Source lines of code (SLOC)	Length	
M2	Cyclomatic complexity (CC)	Complexity	
M3	Maximum nesting depth	Maximum value	
M4	Maximum method chaining	Maximum value	
M5	Unique variable identifiers	Unique count	
M6	Anonymous class declarations	Count	
M7	Arithmetic In- or Decrementations	Count	
M8	Arithmetic infix operations	Count	
M9	Array accesses	Count	
M10	Array creations	Count	
M11	Assignments	Count	
M12	Boolean operators	Count	
M13	Cast expressions	Count	
M14	Catch clauses	Count	
M15	Comparison operators	Count	
M16	If conditions	Count	
M17	Inner method declarations	Count	
M18	Instance-of checks	Count	
M19	Instantiations	Count	
M20	Loops	Count	
M21	Method invocations	Count	
M22	Null checks	Count	
M23	Null literals	Count	
M24	Return statements	Count	
M25	String literals	Count	
M26	Super-method invocations	Count	
M27	Switch-Case blocks	Count	
M28	Synchronized blocks	Count	
M29	Ternary operations	Count	
M30	Throw statements	Count	
M31	Try blocks	Count	
M32	All conditions	Count	
M33	All arithmetic operations	Count	
M34	Is constructor	Boolean	
M35	Is setter	Boolean	
M36	Is getter	Boolean	
M37	Is empty method	Boolean	
M38	Is delegation method	Boolean	
M39	Is ToString method	Boolean	

SLOC is the number of source lines of code, that is, LOC without empty lines and comments. Cyclomatic Complexity corresponds to the metric proposed by McCabe (1976). Despite this metric being controversial (Shepperd, 1988; Hummel, 2014)—due to the fact that it is not actionable, difficult to interpret, and high values do not necessarily translate to low readability—it is commonly used as variable in defect prediction (Menzies et al., 2002, 2004; Zimmermann, Premraj & Zeller, 2007). Furthermore, a low number of paths through a method could be relevant for identifying LFR methods. Maximum nesting depth corresponds to the “maximum number of encapsulated scopes inside the body of the method” (NDepend, 2017). Deeply nested code is more difficult to understand, therefore, it could be more fault-prone. Maximum method chaining expresses the maximum number of chain elements of a method invocation. We consider a method call to be chained if it is directly invoked on the result from the previous method invocation. The value for a method is zero if it does not contain any method invocations, one if no method invocation is chained, or otherwise the maximum number of chain elements (e.g., two for getId().toString(), three for getId().toString().subString(1)). Unique variable identifiers counts the distinct names of variables that are used within the method. The following metrics, metrics M6–M31, count the occurrences of the respective Java language construct (Gosling et al., 2013).

Next, we derive further metrics from the existing ones. They are redundant, but correlated metrics do not have any negative effects on association rule mining (except on the computation time) and may improve the results for the following reason: if an item generated from a metric is not frequent, rules with this item will be discarded because they cannot achieve the minimum support; however, an item for a more general metric may be more frequent and survive. The derived metrics are:All Conditions, which sums up If Conditions, Switch-Case Blocks, and Ternary Operations (M16 + M27 + M29)

All Arithmetic Operations, which sums up Incrementations, Decrementations, and Arithmetic Infix Operations (M7 + M8)

Furthermore, we compute to which of the following categories a method belongs (a method can belong to zero, one, or more categories):Constructors: Special methods that create and initialize an instance of a class. They might be less fault-prone because they often only set class variables or delegate to another constructor.

Getters: Methods that return a class or instance variable. They usually consist of a single statement and can be generated by the IDE.

Setters: Methods that set the value of a class or instance variable. They usually consist of a single statement and can be generated by the IDE.

Empty Methods: Non-abstract methods without any statements. They often exist to meet an implemented interface, or because the default logic is to do nothing and is supposed to be overridden in certain sub-classes.

Delegation Methods: Methods that delegate the call to another method with the same name and further parameters. They often do not contain any logic besides the delegation.

ToString Methods: Implementations of Java’s toString method. They are often used only for debugging purposes and can be generated by the IDE.

Note that we only use source-code metrics and do not consider process metrics. This is because we want to identify methods that exhibit a LFR due to their code.

Association rule mining computes frequent item sets from categorical attributes; therefore, our next step is to discretize the numerical metrics. In defect prediction, discretization is also applied to the metrics: Shivaji et al. (2013) and McCallum & Nigam (1998) reported that binary values can yield better results than using counts when the number of features is low. We discretize as follows:For each of the metrics M1–M5, we inspect their distribution and create three classes. The first class is for metric values until the first tertile, the second class for values until the second tertile, and the third class for the remaining values.

For all count metrics (including the derived ones), we create a binary “has-no”-metric, which is true if the value is zero, e.g., CountLoops = 0 ⇒ NoLoops = true.

For the method categories (setter, getter, …), no transformation is necessary as they are already binary.

Data pre-processing

At this point, we assume that we have a list of faulty methods with their metrics at the faulty state (the list may contain a method multiple times if it was fixed multiple times) and a list of all methods. Faulty methods can be obtained by identifying methods that were changed in bug-fix commits (Zimmermann, Premraj & Zeller, 2007; Giger et al., 2012; Shippey et al., 2016). A method is considered as faulty when it was faulty at least once in its history; otherwise it is considered as not faulty. We describe in “Fault data extraction” how we extracted faulty methods from the Defects4J dataset.

Prior to applying the mining algorithm, we have (1) to address faulty methods with multiple occurrences, (2) to create a unified list of faulty and non-faulty methods, and (3) to tackle dataset imbalance.

Steps (1) and (2) require that a method can be uniquely identified. To satisfy this requirement, we identified a method by its name, its parameter types, and the qualified name of its surrounding class. We integrated the computation of the metrics into the source-code analysis tool Teamscale (Heinemann, Hummel & Steidl, 2014; Haas, Niedermayr & Jurgens, 2019), which is aware of the code history and tracks method genealogies. Thereby, Teamscale detects method renames or parameter changes so that we could update the method identifier when it changed.

A method may be fixed multiple times; in this case, a method appears multiple times in the list of the faulty methods. However, each method should have the same weight and should therefore be considered only once. Consequently, we consolidate multiple occurrences of the same method: we replace all occurrences by a new instance and apply majority voting to aggregate the binary metric values. It is common practice in defect prediction to have a single instance of every method with a flag that indicates whether a method was faulty at least once (Menzies et al., 2010; Giger et al., 2012; Shippey et al., 2016; Mende & Koschke, 2009).

To create a unified dataset, we take the list of all methods, remove those methods that exist in the set of the faulty methods, and add the set of the faulty methods with the metrics computed at the faulty state. After doing that, we end up with a list containing each method exactly once and a flag indicating whether a method was faulty or not.

Defect datasets are often highly imbalanced (Khoshgoftaar, Gao & Seliya, 2010), with faulty methods being underrepresented. Therefore, we apply SMOTE1 , a well-known algorithm for over- and under-sampling, to address imbalance in the dataset used for training (Longadge, Dongre & Malik, 2013; Chawla et al., 2002). It artificially generates new entries of the minority class using the nearest neighbors of these cases and reduces entries from the majority class (Torgo, 2010). If we do not apply SMOTE to highly imbalanced datasets, many non-expressive rules will be generated when most methods are not faulty. For example, if 95% of the methods are not faulty and 90% of them contain a method invocation, rules with high support will be generated that use this association to identify non-faulty methods. Balancing avoids those nonsense rules.

IDP classifier

To identify LFR methods, we compute association rules of the type {Metric1, Metric2, Metric3, …} → {NotFaulty}. Examples for the metrics are SlocLowestThird, NoNullChecks, IsSetter. A method that satisfies all metric predicates of a rule is not faulty to the certainty expressed by the confidence of the rule. The support of the rule expresses how many methods with these characteristics exist, and thus, it shows how generalizable the rule is.

After computing the rules on a training set, we remove redundant ones (see “Association Rule Mining”) and order the remaining rules first descending by their confidence and then by their support. To build the LFR classifier, we combine the top n association rules with the highest confidence values using the logical-or operator. Hence, we consider a method to have a LFR if at least one of the top n rules matches. To determine n, we compute the maximum number of rules until the faulty methods in the LFR methods exceed a certain threshold in the training set.

Of course, IDP can also be used with other machine-learning algorithms. We decided to use association rule mining because of the natural comprehensibility of the rules (see “Association Rule Mining”) and because we achieved a better performance compared to models we trained using Random Forest.

Empirical Study

This section reports on the empirical study that we conducted to evaluate the IDP approach.

Research questions

We investigate the following questions to research how well methods that contain hardly any faults can be identified and to study whether IDP is applicable in cross-project scenarios.

RQ 1: What is the precision of the classifier for low-fault-risk methods? To evaluate the precision of the classifier, we investigate how many methods that are classified as “LFR” (due to the triviality of their code) are faulty. If we want to use the LFR classifier for determining methods that require less focus during quality assurance (QA) activities, such as testing and code reviews, we need to be sure that these methods contain hardly any faults.

RQ 2: How large is the fraction of the code base consisting of methods classified as “low fault risk”? We study how common LFR methods are in code bases to find out how much code is of lower importance for quality-assurance activities. We want to determine which savings potential can arise if these methods are excluded from QA.

RQ 3: Is a trained classifier for methods with low fault risk generalizable to other projects? Cross-project defect prediction is used to predict faults in (new) projects, for which no historical fault data exists, by using models trained on other projects. It is considered a challenging task in defect prediction (He et al., 2012; Zimmermann et al., 2009; Turhan et al., 2009). As we expect that the characteristics of LFR methods might be project-independent, IDP could be applicable in a cross-project scenario. Therefore, we investigate how generalizable our IDP classifier is for cross-project use.

RQ 4: How does the classifier perform compared to a traditional defect prediction approach? The main purpose of defect prediction is to detect fault-prone code. Most traditional defect prediction approaches are binary classifications, which classify a method either as (likely) faulty or not faulty. Hence, they implicitly also detect methods with a LFR. Therefore, we compare the performance of our classifier with the performance of a traditional defect prediction approach.

Study objects

For our analysis, we used data from Defects4J, which was created by Just, Jalali & Ernst (2014). Defects4J is a database and analysis framework that provides real faults for six real-world open-source projects written in Java. For each fault, the original commit before the bug fix (faulty version), the original commit after the bug fix (fixed version), and a minimal patch of the bug fix are provided. The patch is minimal such that it contains only code changes that (1) fix the fault and (2) are necessary to keep the code compilable (e.g., when a bug fix involves method-signature changes). It does not contain changes that do not influence the semantics (e.g., changes in comments, local renamings), and changes that were included in the bug-fix commit but are not related to the actual fault (e.g., refactorings). Due to the manual analysis, this dataset at the method level is much more precise than other datasets at the same level, such as Shippey et al. (2016) and Giger et al. (2012), which were generated from version control systems and issue trackers without further manual filtering. The authors of Just, Jalali & Ernst (2014) confirmed that they considered every bug fix within a given time span.

Table 2 presents the study objects and their characteristics. We computed the metrics SLOC and #Methods for the code revision at the last bug-fix commit of each project; the numbers do not comprise sample and test code. #Faulty methods corresponds to the number of faulty methods derived from the dataset.

Table 2 Study objects.

Name	SLOC	#Methods	#Faulty methods	
JFreeChart (Chart)	81.6k	6.8k	39	
Google Closure compiler	166.7k	13.0k	148	
Apache commons Lang	16.6k	2.0k	73	
Apache Commons Math	9.5k	1.2k	132	
Mockito	28.3k	2.5k	64	
Joda Time	89.0k	10.1k	45	

Fault data extraction

Defects4J provides for each project a set of reverse patches2 , which represent bug fixes. To obtain the list of methods that were at least once faulty, we conducted the following steps for each patch. First, we checked out the source code from the project repository at the original bug-fix commit and stored it as fixed version. Second, we applied the reverse patch to the fixed version to get to the code before the bug fix and stored the resulting faulty version.

Next, we analyzed the two versions created for every patch. For each file that was changed between the faulty and the fixed version, we parsed the source code to identify the methods. We then mapped the code changes to the methods to determine which methods were touched in the bug fix. After that, we had the list of faulty methods. Figure 2 summarizes these steps.

Figure 2 Derivation of faulty methods.

The original bug-fix commit c1e8ed to fix the faulty version f81f3f may contain unrelated changes. Defect4J provides a reverse patch, which contains only the actual fix. We applied it to the fixed version c1e8ed to get to fa30f1 We then identified methods that were touched by the patch and computed their metrics at state fa30f1.

We inspected all 395 bug-fix patches and found that 10 method changes in the patches do not represent bug fixes. While the patches are minimal, such that they contain only bug-related changes (see “Study Objects”), these ten method changes are semantic-preserving, only necessary because of changed signatures of other methods in the patch, and therefore included in Defects4J to keep the code compilable. Figure 3 presents an example. Although these methods are part of the bug fix, they were not changed semantically and do not represent faulty methods. Therefore, we decided to remove them from the faulty methods in our analysis. The names of these ten methods are provided in the dataset to this paper (Niedermayr, Röhm & Wagner, 2019).

Figure 3 Example of method change without behavior modification to preserve API compatibility.

The method escapeJavaScript(String) invokes escapeJavaStyleString(String, boolean, boolean). A further parameter was added to the invoked method; therefore, it was necessary to adjust the invocation in escapeJavaScript(String). For invocations with the parameter value true, the behavior does not change (Lang, patch 46, simplified).

Procedure

After extracting the faulty methods from the dataset, we computed the metrics listed in “IDP Approach.” We computed them for all faulty methods at their faulty version and for all methods of the application code3 at the state of the fixed version of the last patch. We used Eclipse JDT AST (http://www.eclipse.org/jdt/) to create an AST visitor for computing the metrics. For all further processing, we used the statistical computing software R (R Core Team, 2018).

To discretize the metrics M1–M5, we first computed their value distribution. Figure 4 shows that their values are not normally distributed (most values are very small). To create three classes for each of these metrics4 , we sorted the metric values, and computed the values at the end of the first and at the end of the second third. We then put all methods until the last occurrence of the value at the end of the first third into class 1, all methods until the last occurrence of the value at the end of the second third into class 2, and all other methods into class 3. Table 3 presents the value ranges of the resulting classes. The classes are the same for all six projects.

Figure 4 Metrics M1–M5 are not normally distributed.

(A) SLOC, (B) cyclomatic complexity, (C) maximum nesting depth, (D) maximum method chaining, and (E) unique variable identifiers.

Table 3 Generated classes and their value ranges.

Metric	Class 1	Class 2	Class 3	
SLOC	[0;3]	[4;8]	[9;∞)	
Cyclomatic complexity	[0;1]	[2;2]	[3;∞)	
Maximum nesting depth	[0;0]	[1;1]	[2;∞)	
Maximum method chaining	[0;1]	[2;2]	[3;∞)	
Unique variable identifiers	[0;1]	[2;3]	[4;∞)	

We then aggregated multiple faulty occurrences of the same method (this occurs if a method is changed in more than one bug-fix patch) and created a unified dataset of faulty and non-faulty methods (see “Data pre-processing”).

Next, we split the dataset into a training and a test set. For RQ 1 and RQ 2, we used 10-fold cross-validation (Witten et al., 2016, Chapter 5). Using the caret package (from Jed Wing et al. (2017)), we randomly sampled the dataset of each project into ten stratified partitions of equal sizes. Each partition is used once for testing the classifier, which is trained on the remaining nine partitions. To compute the association rules for RQ 3—in which we study how generalizable the classifier is—for each project, we used the methods of the other five projects as training set for the classifier.

Before computing association rules, we applied the SMOTE algorithm from the DMwR package (Torgo, 2010) with a 100% over-sampling and a 200% under-sampling rate to each training set. After that, each training set was equally balanced (50% faulty methods, 50% non-faulty methods)5 .

We then used the implementation of the Apriori algorithm (Agrawal & Srikant, 1994) in the arules package (Hahsler, Gruen & Hornik, 2005; Hahsler et al., 2017) to compute association rules with NotFaulty as target item (rule consequent). We set the threshold for the minimum support to 10% and the threshold for the minimum confidence to 90% (support and confidence are explained in “Association Rule Mining”). We experimented with different thresholds and these values produced good results (results for other configurations are in the dataset provided with this paper; Niedermayr, Röhm & Wagner, 2019). The minimum support avoids overly infrequent (i.e., non-generalizable) rules from being created, and the minimum confidence prevents the creation of imprecise rules. Note that no rule (with NotFaulty as rule consequent) can reach a higher support than 50% after the SMOTE pre-processing. After computing the rules, we removed redundant ones using the corresponding function from the apriori package. We then sorted the remaining rules descending by their confidence.

Using these rules, we created two classifiers to identify LFR methods. They differ in the number of comprised rules. The strict classifier uses the top n rules until the share of faulty methods in all methods (of the training set) exceeds 2.5% in the LFR methods (of the training set). The more lenient classifier uses the top n rules until the share exceeds 5% in the LFR methods. (Example: We applied the top one rule to the training set, then applied the next rule, …, until the matched methods in the training set contained 2.5% out of all faults). Figure 5 presents how an increase in the number of selected rules affects the proportion of LFR methods and the share of faulty methods that they contain. For RQ 1 and RQ 2, the classifiers were computed for each fold of each project. For RQ 3, the classifiers were computed once for each project.

Figure 5 Influence of the number of selected rules (Lang).

The number of rules influences the — proportion of low-fault-risk (LFR) methods and the — share of faulty methods in LFR out of all faulty methods.

To answer RQ 1, we used 10-fold cross-validation to evaluate the classifiers separately for each project. We computed the number and proportion of methods that were classified as “LFR” but contained a fault (≈ false positives). Furthermore, we computed precision and recall. Our main goal is to identify those methods that we can say, with high certainty, contain hardly any faults. Therefore, we consider it as more important to achieve a high precision than to predict all methods that do not contain any faults in the dataset.

As the dataset is imbalanced with faulty methods in the minority, the proportion of faults in LFR methods might not be sufficient to assess the classifiers (SMOTE was applied only to the training set). Therefore, we further computed the fault-density reduction, which describes how much less likely the LFR methods contain a fault. For example, if 40% of all methods are classified as “LFR” and contain 10% of all faults, the factor is 4. It can also be read as: 40% of all methods contain only one fourth of the expected faults. We mathematically define the fault-density reduction factor based on methods as Proportion of LFR methods out of all methodsProportion of faulty LFR methods out of all faulty methods

and based on SLOC as

Proportion of SLOC in LFR methods out of all SLOCProportion of faulty LFR methods out of all faulty methods.

For both classifiers (strict variant with 2.5%, lenient variant with 5%), we present the metrics for each project and the resulting median.

To answer RQ 2, we assessed how common methods classified as “LFR” are. For each project, we computed the absolute number of LFR methods, their proportion out of all methods, and their extent by considering their SLOC. LFR SLOC corresponds to the sum of SLOC of all LFR methods. The proportion of LFR SLOC is computed out of all SLOC of the project.

To answer RQ 3, we computed the association rules for each project with the methods of the other five projects as training data. Like in RQ 1 and RQ 2, we determined the number of used top n rules with the same thresholds (2.5% and 5%). To allow a comparison with the within-project classifiers, we computed the same metrics like in RQ 1 and RQ 2.

To answer RQ 4, we computed for each method the 9 code and 15 change metrics that were used in Giger et al. (2012). The metrics and their descriptions are listed in Table 4. We applied Random Forest as machine learning algorithm and configured it like in the paper of Giger et al. (2012). We computed the results for within-project predictions using 10-fold cross-validation and we further computed the results for cross-project predictions like in RQ 3. We present the same evaluation metrics as in the previous research questions.

Table 4 RQ 4: Code and change metrics used in Giger et al. (2012).

Metric name	Description	
Code metrics	
fanIN	Number of methods that reference a given method	
fanOUT	Number of methods referenced by a given method	
localVar	Number of local variables in the body of a method	
parameters	Number of parameters in the declaration	
commentToCodeRatio	Ratio of comments to source code (line-based)	
countPath	Number of possible paths in the body of a method	
complexity	McCabe cyclomatic complexity of a method	
execStmt	Number of executable source code statements	
maxNesting	Maximum nested depth of all control structures	
Change metrics	
methodHistories	Number of times a method was changed	
authors	Number of distinct authors that changed a method	
stmtAdded	Sum of all source code statements added to a method body	
maxStmtAdded	Maximum number of source code statements added to a method body	
avgStmtAdded	Average number of source code statements added to a method body	
stmtDeleted	Sum of all source code statements deleted from a method body	
maxStmtDeleted	Maximum number of source code statements deleted from a method body	
avgStmtDeleted	Average number of source code statements deleted from a method body	
churn	Sum of churn (stmtAdded—stmtDeleted)	
maxChurn	Maximum churn	
avgChurn	Average churn	
decl	Number of method declaration changes	
cond	Number of condition expression changes in a method body	
elseAdded	Number of added else-parts in a method body	
elseDeleted	Number of deleted else-parts from a method body	

Results

This section presents the results to the research questions. The data to reproduce the results is available at (Niedermayr, Röhm & Wagner, 2019).

RQ 1: What is the precision of the classifier for low-fault-risk methods? Table 5 presents the results. The methods classified to have LFR by the stricter classifier, which allows a maximum fault share of 2.5% in the LFR methods in the (balanced) training data, contain between two and eight faulty methods per project. The more lenient classifier, which allows a maximum fault share of 5%, classified between four and 15 faulty methods as LFR. The median proportion of faulty methods in LFR methods is 0.3% resp. 0.4%.

Table 5 RQ 1, RQ 2: Evaluation of within-project IDP to identify low-fault-risk (LFR) methods.

Project	Faults in LFR	LFR methods	LFR methods	LFR SLOC	LFR methods contain …% of all faults	Fault-density reduction	
#	%	Prec	Rec	#	%	#	%	(Methods)	(SLOC)	
Within-project IDP, 10-fold: min. support = 10%, min. confidence = 90%, rules until fault share in training set = 2.5%	
Chart	4	0.1%	99.9%	44.1%	2,995	43.9%	11,228	15.8%	10.3%	4.3	1.5	
Closure	6	0.2%	99.8%	29.2%	3,759	28.9%	15,497	10.5%	4.1%	7.1	2.6	
Lang	3	0.5%	99.5%	29.6%	576	28.6%	2,242	13.8%	4.1%	7.0	3.4	
Math	2	1.1%	98.9%	18.4%	190	16.5%	570	4.8%	1.5%	10.9	3.1	
Mockito	5	0.6%	99.4%	35.1%	875	34.4%	6,128	25.1%	7.8%	4.4	3.2	
Time	8	0.1%	99.9%	80.4%	8,063	80.2%	62,063	78.1%	17.8%	4.5	4.4	
Median		0.3%	99.7%	32.3%		31.7%		14.8%	6.0%	5.7	3.2	
Within-project IDP, 10-fold: min. support = 10%, min. confidence = 90%, rules until fault share in training set = 5%	
Chart	4	0.1%	99.9%	44.8%	3,040	44.6%	11,563	16.3%	10.3%	4.3	1.6	
Closure	15	0.3%	99.7%	41.8%	5,385	41.5%	25,981	17.6%	10.1%	4.1	1.7	
Lang	6	0.7%	99.3%	45.0%	879	43.7%	3,630	22.3%	8.2%	5.3	2.7	
Math	7	2.7%	97.3%	24.3%	255	22.1%	878	7.3%	5.3%	4.2	1.4	
Mockito	6	0.5%	99.5%	47.8%	1,189	46.8%	8,260	33.8%	9.4%	5.0	3.6	
Time	9	0.1%	99.9%	82.8%	8,298	82.5%	63,333	79.7%	20.0%	4.1	4.0	
Median		0.4%	99.6%	44.9%		44.1%		20.0%	9.8%	4.3	2.2	

The fault-density reduction factor for the stricter classifier ranges between 4.3 and 10.9 (median: 5.7) when considering methods and between 1.5 and 4.4 (median: 3.2) when considering SLOC. In the project Lang, 28.6% of all methods with 13.8% of the SLOC are classified as LFR and contain 4.1% of all faults, thus, the factor is 7.0 (SLOC-based: 3.4). The factor never falls below 1 for both classifiers.

IDP can identify methods with LFR. On average, only 0.3% of the methods classified as “LFR” by the strict classifier are faulty. The identified LFR methods are, on average, 5.7 times less likely to contain a fault than an arbitrary method in the dataset.

Table 6 exemplarily presents the top three rules for Lang. Methods that work with fewer than two variables and do not invoke any methods as well as short methods without arithmetic operations, cast expressions, and method invocations are highly unlikely to contain a fault.

Table 6 Top three association rules for Lang (within-project, fold 1).

#	Rule	Support (%)	Confidence (%)	
1	{UniqueVariableIdentifiersLessThan2, NoMethodInvocations} ⇒ {NotFaulty}	10.98	100.00	
2	{SlocLessThan4, NoMethodInvocations, NoArithmeticOperations} ⇒ {NotFaulty}	10.98	100.00	
3	{SlocLessThan4, NoMethodInvocations, NoCastExpressions} ⇒ {NotFaulty}	10.60	100.00	

RQ 2: How large is the fraction of the code base consisting of methods classified as “low fault risk”? Table 5 presents the results. The stricter classifier classified between 16.5% and 80.2% of the methods as LFR (median: 31.7%, mean: 38.8%), the more lenient classifier matched between 22.1% and 82.5% of the methods (median: 44.1%, mean: 46.9%). The median of the comprised SLOC in LFR methods is 14.8% (mean: 24.7%) respectively 20.0% (mean: 29.5%).

Using within-project IDP, on average, 32–44% of the methods, comprising about 15–20% of the SLOC, can be assigned a lower importance during testing.

In the best case, when ignoring 16.5% of the methods (4.8% of the SLOC), it is still possible to catch 98.5% of the faults (Math).

RQ 3: Is a trained classifier for methods with low fault risk generalizable to other projects? Table 7 presents the results for the cross-project prediction with training data from the respective other projects. Compared to the results of the within-project prediction, except for Math, the number of faults in LFR methods decreased or stayed the same in all projects for both classifier variants. While the median proportion of faults in LFR methods slightly decreased, the proportion of LFR methods also decreased in all projects except Math. The median proportion of LFR methods is 23.3% (SLOC: 8.1%) for the stricter classifier and 26.3% (SLOC: 12.6%) for the more lenient classifier.

Table 7 RQ 3: Evaluation of cross-project IDP.

Project	Faults in LFR	LFR methods	LFR methods	LFR SLOC	LFR methods contain …% of all faults	Fault-density reduction	
#	%	Prec.	Rec.	#	%	#	%	(Methods)	(SLOC)	
Cross-project IDP: min. support = 10%, min. confidence = 90%, rules until fault share in training set = 2.5%	
Chart	3	0.1%	99.9%	32.1%	2,182	32.0%	7,434	10.5%	7.7%	4.2	1.4	
Closure	2	0.1%	99.9%	25.0%	3,207	24.7%	11,584	7.9%	1.4%	18.3	5.8	
Lang	1	0.2%	99.8%	23.1%	449	22.3%	1,357	8.3%	1.4%	16.3	6.1	
Math	8	2.9%	97.1%	26.6%	280	24.3%	1,129	9.4%	6.1%	4.0	1.6	
Mockito	1	0.2%	99.8%	21.7%	539	21.2%	1,698	6.9%	1.6%	13.6	4.4	
Time	1	0.1%	99.9%	18.4%	1,845	18.3%	5,807	7.3%	2.2%	8.3	3.3	
Median		0.2%	99.8%	24.0%		23.3%		8.1%	1.9%	10.9	3.9	
Cross-project IDP: min. support = 10%, min. confidence = 90%, rules until fault share in training set = 5%	
Chart	4	0.2%	99.8%	35.5%	2,411	35.4%	9,363	13.2%	10.3%	3.4	1.3	
Closure	4	0.1%	99.9%	25.9%	3,327	25.6%	15,583	10.6%	2.7%	9.5	3.9	
Lang	4	0.7%	99.3%	27.7%	542	26.9%	1,959	12.0%	5.5%	4.9	2.2	
Math	18	5.1%	94.9%	32.9%	354	30.7%	1,634	13.7%	13.6%	2.2	1.0	
Mockito	1	0.2%	99.8%	25.0%	620	24.4%	3,495	14.3%	1.6%	15.6	9.1	
Time	1	0.0%	100.0%	20.0%	2,007	20.0%	7,552	9.5%	2.2%	9.0	4.3	
Median		0.2%	99.8%	26.8%		26.3%		12.6%	4.1%	6.9	3.1	

The fault-density reduction improved compared to the within-project prediction for both the method and SLOC level in both classifier variants: For the stricter classifier, the median of the method-based factor is 10.9 (+5.2); the median of the SLOC-based factor is 3.9 (+0.7). Figure 6 illustrates the fault-density reduction for both within-project (RQ 1, RQ 2) and cross-project (RQ 3) prediction.

Figure 6 Comparison of the IDP within-project ( 2.5%,  5.0%) with the IDP cross-project ( 2.5%,  5.0%) classifiers (method-based).

The fault-density reduction expresses how much less likely a LFR method contains a fault (definition in Procedure). Higher values are better (Example: If 40% of the methods are LFR and contain 5% of all faults, the factor is 8). The dashed line is at one; no value falls below.

Using cross-project IDP, on average, 23–26% of the methods, comprising about 8–13% of the SLOC, can be classified as “LFR.” The methods classified by the stricter classifier contain, on average, less than one eleventh of the expected faults.

RQ 4: How does the classifier perform compared to a traditional defect prediction approach? Table 8 presents the results of the within- and cross-project prediction according to the approach by Giger et al. (2012). In the within-project prediction scenario, the classifier predicts on average 99.4% of the methods to be non-faulty. As a consequence, the average recall regarding non-faulty methods reaches 99.9%. However, the number of methods that are classified as non-faulty but actually contain a fault increases by magnitudes compared to the IDP approach (i.e., precision deteriorates). For example, 77% of Closure’s faulty methods are wrongly classified as non-faulty. The median fault-density reduction is 1.6 at the method level (strict IDP: 5.7) and 1.4 when considering SLOC (strict IDP: 3.2). Consequently, methods classified by the traditional approach to have a LFR are still less likely to contain a fault than other methods, but the difference is not as high as in the IDP classifier.

Table 8 RQ 4: Results of a traditional defect prediction approach.

Project	Faults in LFR	LFR methods	LFR methods	LFR SLOC	LFR methods contain …% of all faults	Fault-density reduction	
#	%	Prec.	Rec	#	%	#	%	(Methods)	(SLOC)	
Within-project defect prediction: traditional approach used in Giger et al. (2012)	
Chart	36	0.5%	99.5%	99.9%	6,785	99.9%	70,457	99.6%	92.3%	1.1	1.1	
Closure	114	0.9%	99.1%	99.9%	12,862	99.6%	142,518	97.6%	77.0%	1.3	1.3	
Lang	36	1.9%	98.1%	99.5%	1,905	97.6%	14,462	91.2%	49.3%	2.0	1.9	
Math	62	6.4%	93.6%	98.4%	967	91.9%	7,234	66.5%	47.0%	2.0	1.4	
Mockito	46	1.9%	98.1%	99.8%	2,481	99.1%	24,232	99.5%	71.9%	1.4	1.4	
Time	26	0.3%	99.7%	100.0%	9,995	99.8%	78,809	99.8%	57.8%	1.7	1.7	
Median		1.4%	98.6%	99.9%		99.4%		98.6%	64.8%	1.6	1.4	
Cross-project defect prediction: traditional approach used in Giger et al. (2012)	
Chart	32	0.5%	99.5%	99.2%	6,755	99.1%	68,214	96.3%	82.1%	1.2	1.2	
Closure	82	0.6%	99.4%	99.5%	12,859	99.0%	141,322	96.0%	55.4%	1.8	1.7	
Lang	52	2.6%	97.4%	99.9%	1,990	98.9%	15,085	92.8%	71.2%	1.4	1.3	
Math	103	9.2%	90.8%	99.8%	1,123	97.3%	9,512	79.5%	78.0%	1.2	1.0	
Mockito	58	2.3%	97.7%	100.0%	2,534	99.8%	24,420	99.8%	90.6%	1.1	1.1	
Time	39	0.4%	99.6%	100.0%	10,050	99.9%	79,191	99.7%	86.7%	1.2	1.2	
Median		1.5%	98.5%	99.9%		99.0%		96.1%	80.0%	1.2	1.2	

The results in the cross-project prediction scenario are similar. In four of the six projects, the number of faults in LFR methods increased compared with the within-project prediction scenario. The fault-density reduction deteriorated to 1.2 both at the method and SLOC level (strict IDP: 10.9 resp. 3.9). For all projects, IDP outperformed the traditional approach.

Discussion

The results of our empirical study show that only very few LFR methods actually contain a fault, and thus, they indicate that IDP can successfully identify methods that are not fault-prone. On average, 31.7% of the methods (14.8% of the SLOC) matched by the strict classifier contain only 6.0% of all faults, resulting in a considerable fault-density reduction for the matched methods. In any case, LFR methods are less fault-prone than other methods (fault-density reduction is higher than one in all projects); based on methods, LFR methods are at least twice less likely to contain a fault. For the stricter classifier, the extent of the matched methods, which could be deferred in testing, is between 5% and 78% of the SLOC of the respective project. The more lenient classifier matches more methods and SLOC at the cost of a higher fault proportion, but still achieves satisfactory fault-density reduction values. This shows that the balance between fault risk and matched extent can be influenced by the number of considered rules to reflect the priorities of a software project.

Interestingly, the cross-project IDP classifier, which is trained on data from the respective other five projects, exhibits a higher precision than the within-project IDP classifier. Except for the Math project, the LFR methods contain fewer faulty methods in the cross-project prediction scenario. This is in line with the method-based fault-density reduction factor of the strict classifier, which is in four of six cases better in the cross-project scenario (SLOC-based: three of six cases). However, the proportion of matched methods decreased compared to the within-project prediction in most projects. Accordingly, the cross-project results suggest that a larger, more diversified training set identifies LFR methods more conservatively, resulting in a higher precision and lower matching extent.

Math is the only project in which IDP within-project prediction outperformed IDP cross-project prediction. This project contains many methods with mathematical computations expressed by arithmetic operations, which are often wrapped in loops or conditions; most of the faults are located in these methods. Therefore, the within-project classifiers used few, very precise rules for the identification of LFR methods.

To sum up, our results show that the IDP approach can be used to identify methods that are, due to the “triviality” of their code, less likely to contain any faults. Hence, these methods require less focus during quality-assurance activities. Depending on the criticality of the system and the risk one is willing to take, the development of tests for these methods can be deferred or even omitted in case of insufficient available test resources. The results suggest that IDP is also applicable in cross-project prediction scenarios, indicating that characteristics of LFR methods differ less between projects than characteristics of faulty methods do. Therefore, IDP can be used in (new) projects with no (precise) historical fault data to prioritize the code to be tested.

Limitations

A limitation of IDP is that even LFR methods can contain faults. An inspection of faulty methods incorrectly classified to have a LFR showed that some faults were fixed by only adding further statements (e.g., to handle special cases). This means that a method can be faulty even if the existing code as such is not faulty (due to missing code). Further imaginable examples for faulty LFR methods are simple getters that return the wrong variable, or empty methods that are unintentionally empty. Therefore, while these methods are much less fault-prone, it cannot be assumed that they never contain any fault. Consequently, excluding LFR methods from testing and other QA activities carries a risk that needs to be kept in mind.

Relation to defect prediction

As discussed in detail in “Introduction,” IDP presents another view on defect prediction. The focus of IDP on LFR methods requires an optimization toward precision, so that hardly any faulty methods are erroneously classified as trivial. The comparison with a traditional defect prediction approach showed that IDP classified much fewer methods as trivial. However, methods classified by IDP as trivial contain far fewer faulty methods, that is, IDP achieves a higher precision. Consequently, the identified trivial methods can be deferred or even excluded from quality-assurance activities.

Threats to validity

Next, we discuss the threats to internal and external validity.

Threats to internal validity

The learning and evaluation was performed on information extracted from Defects4J (Just, Jalali & Ernst, 2014). Therefore, the quality of our data depends on the quality of Defects4J. Common problems for defect datasets created by analyzing changes in commits that reference a bug ticket in an issue tracking system are as follows. First, commits that fix a fault but do not reference a ticket in the commit message cannot be detected (Bachmann et al., 2010). Consequently, the set of commits that reference a bug fix may not be a fair representation of all faults (Bird et al., 2009; D’Ambros, Lanza & Robbes, 2012; Giger et al., 2012). Second, bug tickets in the issue tracker may not always represent faults and vice versa. Herzig, Just & Zeller (2013) pointed out that a significant amount of tickets in the issue trackers of open-source projects is misclassified. Therefore, it is possible that not all bug-fix commits were spotted. Third, methods may contain faults that have not been detected or fixed yet. In general, it is not possible to prove that a method does not contain any faults. Fourth, a commit may contain changes (such as refactorings) that are not related to the bug fix, but this problem does not affect the Defects4J dataset due to the authors’ manual inspection. These threats are present in nearly all defect prediction studies, especially in those operating at the method level. Defect prediction models were found to be resistant to such kind of noise to a certain extent (Kim et al., 2011).

Defects4J contains only faults that are reproducible and can be precisely mapped to methods; therefore, faulty methods may be under-approximated. In contrast, other datasets created without manual post-processing tend to over-approximate faults. To mitigate this threat, we replicated our IDP evaluation with two study objects used in Giger et al. (2012). The observed results were similar to our study.

Threats to external validity

The empirical study was performed with six mature open-source projects written in Java. The projects are libraries and their results may not be applicable to other application types, for example, large industrial systems with user interfaces. The results may also not be transferable to projects of other languages, for the following reasons: First, Java is a strongly typed language that provides type safety. It is unclear if the IDP approach works for languages without type safety, because it could be that even simple methods in such languages exhibit a considerable amount of faults. Second, in case the approach as such is applicable to other languages, the collected metrics and the LFR classifier need to be validated and adjusted. Other languages may use language constructs in a different way or use constructs that do not exist in Java. For example, a classifier for the C language should take constructs such as GOTOs and the use of pointer arithmetic into consideration. Furthermore, the projects in the dataset (published in 2014) did not contain code with lambda expressions introduced in Java 8 (http://www.oracle.com/technetwork/articles/java/architect-lambdas-part1-2080972.html). Therefore, in newer projects that make use of lambda expressions, the presence of lambdas should be taken into consideration when classifying methods. Consequently, further studies are necessary to determine whether the results are generalizable.

Like in most defect prediction studies, we treated all faults as equal and did not consider their severity. According to Ostrand, Weyuker & Bell (2004), the severity of bug tickets is often highly subjective. In reality, not all faults have the same importance, because some cause higher failure follow-up costs than others.

Conclusion

Developer teams often face the problem of scarce test resources and need therefore to prioritize their testing efforts (e.g., when writing new automated unit tests). Defect prediction can support developers in this activity. In this paper, we propose an inverse view on defect prediction (IDP) to identify methods that are so “trivial” that they contain hardly any faults. We study how unerringly such LFR methods can be identified, how common they are, and whether the proposed approach is applicable for cross-project predictions.

We show that IDP using association rule mining on code metrics can successfully identify LFR methods. The identified methods contain considerably fewer faults than the average code and can provide a savings potential for QA activities. Depending on the parameters, a lower priority for QA can be assigned on average to 31.7% resp. 44.1% of the methods, amounting to 14.8% resp. 20.0% of the SLOC. While cross-project defect prediction is a challenging task (He et al., 2012; Zimmermann et al., 2009), our results suggest that the IDP approach can be applied in a cross-project prediction scenario at the method level. In other words, an IDP classifier trained on one or more (Java open-source) projects can successfully identify LFR methods in other Java projects for which no—or no precise—fault data exists.

For future work, we want to replicate this study with closed-source projects, projects of other application types, and projects in other programming languages. It is also of interest to investigate which metrics and classifiers are most effective for the IDP purpose and whether they differ from the ones used in traditional defect prediction. Moreover, we plan to study whether code coverage of LFR methods differs from code coverage of other methods. If guidelines to meet a certain code coverage level are set by the management, unmotivated testers may add tests for LFR methods first because it might be easier to write tests for those methods. Consequently, more complex methods with a higher fault risk may remain untested once the target coverage is achieved. Therefore, we want to investigate whether this is a problem in industry and whether it can be addressed with an adjusted code-coverage computation, which takes LFR methods into account.

We thank Nils Göde, Florian Deißenböck, and the anonymous reviewers for their valuable feedback.

Additional Information and Declarations

Competing Interests

Author Contributions

Data Availability

1 Synthetic minority over-sampling technique.

2 A reverse patch reverts previous changes.

3 Code without sample and test code.

4 We did not use the ntile function to create classes, because it always generates classes of the same size, such that instances with the same value may end up in different classes (e.g., if 50% of the methods have the complexity value 1, the first 33.3% will end up in class 1, and the remaining 16.7% with the same value will end up in class 2).

5 We computed the results for the empirical study once with and once without addressing the data imbalance in the training set. The prediction performance was better when applying SMOTE, therefore, we decided to use it.

Rainer Niedermayr and Tobias Röhm are employees of CQSE GmbH. The responsibility for this article lies with the authors.

Rainer Niedermayr conceived and designed the experiments, performed the experiments, analyzed the data, contributed reagents/materials/analysis tools, prepared figures and/or tables, performed the computation work, authored or reviewed drafts of the paper, approved the final draft.

Tobias Röhm conceived and designed the experiments, analyzed the data, authored or reviewed drafts of the paper, approved the final draft.

Stefan Wagner conceived and designed the experiments, analyzed the data, authored or reviewed drafts of the paper, approved the final draft.

The following information was supplied regarding data availability:

Niedermayr, Rainer (2019): Dataset: Too Trivial To Test? figshare. Fileset. DOI 10.6084/m9.figshare.6194171.v1.

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
