# Peer review of "Too trivial to test? An inverse view on defect prediction to identify methods with low fault risk"

_PeerJ Computer Science, doi:10.7717/peerj-cs.187_

## Round 0.1 · original submission · Major Revisions

Please address carefully the reviewers comments. In particular the observation of reviewer 2 since they provide several pointers that can substantially improve the overall quality of the manuscript.

Reviewer 1 ·

Basic reporting

See my general comments

Experimental design

See my general comments

Validity of the findings

See my general comments

Additional comments

The authors propose an approach to identify methods that are less likely to be buggy. Their work has different stakeholders ranging from developers who can minimize their test efforts to researchers who can improve their prediction model, not only within a single application. But, also for cross-project prediction models.

The authors base their approach on mining association rules. It consists of mapping a set of 39 source code metrics to the dependent variable that defines if a method is faulty or not. By this approach, the authors established a set of rules, each of which has support higher than 20% and a confidence level lower than 90%.

On six projects from the Defects4J dataset, the authors validated their model using ten cross-validation technique, while they studied the performance of their models on other projects using five projects to build the model and one project as a testing data. The authors were able to achieve high performance.

The strengths of the paper:
- Very nice idea.
- The article is well written and motivated.
- A solid approach that is well enforced by existing research work.
- Interesting findings.

Weaknesses:

It is not 100% clear if a low-risk method is a method that never had a bug in the past, or if the authors consider fixed methods to be correct methods. The authors said that "First, we checked out the source code from the project repository at the original bug-fix commit and stored it as fixed version." It is not clear if they consider one method through the whole history as buggy or not, or if they can consider one fixed method as a correct method, while they consider it as a buggy method before the fix.

That opens another concern how the authors know if a fixed method doesn't contain another bug, which might be fixed in the future?

While the number of independent metrics used in the paper is significant, building rules from high-level metrics would also be interesting. Instead of the number of try blocks, throw statements, catch clauses, it would also be interesting to define a category that encapsulates all of these in one group such as exception. Then, study the impact of these groups so that readers can know the effect of a practice instead of just primary metrics.

The formulation of RQ1 is very confusing, while it is just evaluating the precision of the model. Simplifying it would be interesting.

Figure 3 shows a modification in the API of a method. So, how do you distinguish between methods? Do you consider just method names? Names and parameters? That is because one can define multiple methods with the same name and different parameters. From the paper, it is not clear if the method (as an example) in Figure 3 becomes another method or not.

Discuss how do you count the confidence and support.

Reviewer 2 ·

Basic reporting

Overall the paper is well written and easy to read. The case study is designed and conducted rigorously. The authors also thoroughly discussed the threats to validity from different angles.

Experimental design

The author claims repeatedly in the paper that recall do not matter but I do not find their argument convincing, In fact, the larger the number of low-risk methods retrieved, the smaller the number of methods that are left to be carefully inspected. Which would result in lower testing resources spent!

Validity of the findings

To justify the need of training classifiers specifically for the detection of trivial methods, I believe that the author should compare the obtained results with those of traditional bug prediction approaches where methods with low probabilities of containing a fault are considered “trivial to test”. In fact, many traditional bug prediction approaches are binary classification approaches, that focus both on detecting code with high and low risk of fault. Hence, they can also be used to detect methods with low risk of fault. A thorough comparison between these traditional approaches and the proposed approach is necessary.

-Also, it would have been interesting to see some assessment of the effort required from developers to examine the identified ‘trivial’ methods. It is very possible that these trivial methods are also very easily identified by developers as low risk (e.g., empty methods, getters, and setters).

Additional comments

This paper proposes an inverse view on software defect prediction, where the idea is to identify methods unlikely to contain faults, rather than those likely to contain faults. The authors extracted metrics from the source code of the projects in Defects4J database and used the Apriori algorithm to create rules to identify non-faulty methods. Through a case study with six Java open-source projects containing precise fault data at the method level. The authors show that their proposed approach can identify between 32% and 44% of the methods of a project to have a low fault risk. These methods are about six times less likely to contain a fault compared to other methods in the studied database. In cross-project predictions, the identified methods are eleven times less likely to contain a fault.


+ Overall the paper is well written and easy to read. The case study is designed and conducted rigorously. The authors also thoroughly discussed the threats to validity from different angles.

Although I like the idea of the paper, I have some doubts about its usefulness for the following reasons:

-The reported recall values are quite low in general (e.g., 18.4% for Math), which mean that many of the “trivial” methods are missed, which is detrimental to the objective of reduction of the testing effort. The author claims repeatedly in the paper that recall do not matter but I do not find their argument convincing, In fact, the larger the number of low-risk methods retrieved, the smaller the number of methods that are left to be carefully inspected. Which would result in lower testing resources spent!

- The paper reports that “28.6% of all methods with 13.8% of the SLOC are classified as LFR and contain 4.1% of all faults”. I wonder what is the severity of these faults that are contained in the false positive? The authors briefly discussed them in Section 6.1, but I believe that a more detailed investigation is needed.

-Also, to justify the need of training classifiers specifically for the detection of trivial methods, I believe that the author should compare the obtained results with those of traditional bug prediction approaches where methods with low probabilities of containing a fault are considered “trivial to test”. In fact, many traditional bug prediction approaches are binary classification approaches, that focus both on detecting code with high and low risk of fault. Hence, they can also be used to detect methods with low risk of fault. A thorough comparison between these traditional approaches and the proposed approach is necessary.

-Also, it would have been interesting to see some assessment of the effort required from developers to examine the identified ‘trivial’ methods. It is very possible that these trivial methods are also very easily identified by developers as low risk (e.g., empty methods, getters, and setters).

---

## Round 0.2 · accepted · Accept

As you can read, the Reviewer is satisfied with the paper revision since it addresses almost all of its concerns.

Please perform the last proofreading of the paper carefully in order to ensure the high-quality PeerJ CS deserves for the published works.

Reviewer 2 ·

Basic reporting

The paper is well written in general.

Experimental design

Experiments are rigorous in general.

Validity of the findings

The reported results are valid. Thank you to the authors for clarifying the context in which they are expected to be used.

Additional comments

I would like to thank the authors for the revisions and the detailed response. I am happy to recommend the acceptance of the paper.